# HM-Chromanone Ameliorates Hyperglycemia and Dyslipidemia in Type 2 Diabetic Mice

**DOI:** 10.3390/nu14091951

**Published:** 2022-05-06

**Authors:** Jae Eun Park, Jaemin Son, Youngwan Seo, Ji Sook Han

**Affiliations:** 1Department of Food Science and Nutrition, Pusan National University, Busan 46241, Korea; jaeeun5609@naver.com; 2Division of Marine Bioscience, Ocean Science & Technology School, Korea Maritime and Ocean University, Busan 49112, Korea; rh7978@naver.com (J.S.); ywseo@hhu.ac.kr (Y.S.)

**Keywords:** HM-chromanone, high blood glucose, AMPK, type 2 diabetes, db/db mice

## Abstract

The effects of (*E*)-5-hydroxy-7-methoxy-3-(2-hydroxybenzyl)-4-chromanone (HMC) on hyperglycemia and dyslipidemia were investigated in diabetic mice. Mice were separated into three groups: db/db, rosiglitazone and HMC. Blood glucose or glycosylated hemoglobin values in HMC-treated mice were significantly lower compared to db/db mice. Total cholesterol, LDL-cholesterol, and triglyceride values were lower, and HDL-C levels were higher, in the HMC group compared to the diabetic and rosiglitazone groups. HMC markedly increased IRS-1^Tyr612^, Akt^Ser473^ and PI3K levels and plasma membrane GLUT4 levels in skeletal muscle, suggesting improved insulin resistance. HMC also significantly stimulated AMPK^Thr172^ and PPARα in the liver, and ameliorated dyslipidemia by inhibiting SREBP-1c and FAS. Consequently, HMC reduced hyperglycemia by improving the expression of insulin-resistance-related genes and improved dyslipidemia by regulating fatty acid synthase and oxidation-related genes in db/db mice. Therefore, HMC could ameliorate hyperglycemia and dyslipidemia in type 2 diabetic mice.

## 1. Introduction

High blood glucose can impair insulin signaling in insulin target tissues, exacerbating a damaged glucose uptake system [1,2,3,4]. To treat type 2 diabetes mellitus (T2DM), it is important to alleviate hyperglycemia. It commonly regulates glucose homeostasis in liver and skeletal muscle, which use most of the available glucose [5,6]. T2DM can cause dyslipidemia by raising levels of triglycerides (TG) and total cholesterol (TC) and lowering levels of high-density lipoprotein (HDL)-cholesterol in the blood [7,8]. Additionally, increased visceral triglyceride stores increase the transport of free fatty acids to the liver [9,10]. Hepatic lipid accumulation may be an important point in insulin resistance [11]. Therefore, the management of high blood glucose and hepatic dyslipidemia is important for reducing the hazard of diabetes-related side effects [12].

The two pathways responsible for glucose uptake in skeletal muscle are the PI3K/Akt and AMPK pathways. Insulin promotes glucose uptake by increasing the translocation of GLUT4 from an intracellular pool to the plasma membrane through activation of the PI3K/Akt pathway [5]. Glucose uptake is also stimulated by AMPK, an intracellular energy sensor that plays an important role in regulating glucose and lipid metabolism [13,14]. The AMPK pathway decreases fatty acid synthesis by inhibiting SREBP-1c. SREBP-1c controls hepatic fatty acid and triacylglycerol biosynthesis by regulating the expression of key genes, such as FAS [15,16]. Thus, the inhibition of these genes through AMPK activation could suppress lipogenesis and improve dyslipidemia.

A lot of research is steadily progressing toward the development of safe and efficacious natural products to overcome the limitations and adverse effects of synthetic drugs [17,18]. HMC is a sappanin homoisoflavonoid, in which benzopyran and aromatic rings are connected through single carbon. To the best of our knowledge, no study has assessed the effects of HMC on C57BL/KsJ-db/db mice. In this model, diabetes-related metabolic changes, including hyperglycemia, hyperinsulinemia, dyslipidemia, and insulin resistance, can be maintained [19,20,21]. Thus, the db/db mouse provides an appropriate model for investigating the effects of HMC on metabolic changes and molecular targets in type 2 diabetes. This study investigated the effects of HMC supplementation on hyperglycemia and dyslipidemia in diabetic mice.

## 2. Materials and Methods

### 2.1. Preparation of Materials

Isolation of HMC from *P. oleracea* was performed in our laboratory with a previously established method [22]. The constitutional formula of HMC is shown in Figure 1.

### 2.2. Animals

C57BL/KsJ-db/db mice (♂, 5 weeks) were obtained from JOONG AH BIO (Suwon-si, Gyeonggi-do, Korea). Mice were fed a commercial diet for 2 weeks. At 7 weeks old, the animals were randomly separated into three groups (*n* = 8): db/db, db/db-RG, and db/db-HMC groups. RG and HMC were dissolved in distilled water. The RG and HMC group were orally administered rosiglitazone (RG, Sigma, St. Louis, MO, USA) and HMC (30 mg/kg BW (body weight) per day) by gavage, once per day, for 6 weeks. After 6 weeks, animals were narcotized by CO_2_ after a 12 h fasting time, and blood was collected from the inferior vena cava for biomarkers experiments. Plasma and tissues (liver and skeletal muscle tissue) were collected and stored in deep freezer. All animal experiments were performed in accordance with the guidelines for management and use of laboratory animals at Pusan National University (PNU-2019-2196).

### 2.3. Blood Glucose and Glycosylated Hemoglobin Levels

Every week, the glucose levels in blood from tail veins were measured after a 12 h fasting time by glucometer (Roche Diagnostics GmbH, Mannheim, Germany). Anticoagulated whole-blood samples were hemolyzed, and glycosylated hemoglobin (HbA1c) was determined by immunoturbidimetry.

### 2.4. Plasma Insulin Level

Plasma insulin levels were measured using a radioimmunoassay with ELISA kit (Linco Research, Inc., Billerica, MA, USA).

### 2.5. Homeostatic Index of Insulin Resistance

HOMA-IR values were computed by the homeostasis model using the following formula: HOMA-IR = {Fasting glucose (mmol/L) × fasting insulin (IU/Ml)}/22.51.

### 2.6. Serum Lipid Levels

The serum was separated, and the lipid parameters were determined. Serum TC, TG, HDL-cholesterol, and LDL-cholesterol levels were determined by Asan kit (Asan Pharmaceutical Corp., Seoul, Korea).

### 2.7. Plasma Membrane Fraction of Skeletal Muscle

Muscle membranes subfractionation was performed as described by Baron et al. [23] using a process adapted by Klip et al. [24]. Skeletal muscle was homogenized and centrifuged at 1000× *g* for ten minutes, and the supernatant was collected. Pellets were resuspended in a buffer and rehomogenized in a glass tube. The supernatants were gathered, mixed with previous supernatant, and the mixture was centrifuged. The resulting supernatant was centrifuged at 190,000× *g* for 1 h.

### 2.8. Western Blot Analysis

Next, 20 μL of protein sample was separated on 12% Tris-HCL resolving gels and transferred to nitrocellulose membranes. Membrane was blocked with 5% skim milk in Tris-buffer and 0.1% Tween20 for 60 min. The blocked membrane was incubated with the respective antibodies for 60 min. Membrane was rinsed and incubated with a secondary antibody for 60 min. Antigen–antibody mixture was visualized by chemiluminescence detection reagents. Chemiluminescence signal was identified by a LAS-1000 plus instrument (Fujifilm, Tokyo, Japan). Band density was measured by Multi Gauge V3.1 (Fujifilm, Tokyo, Japan).

### 2.9. Statistical Analyses

Statistical analysis was measured by SPSS (IBM Corp., Armonk, NY, USA). Differences between the groups were evaluated for significance by one-way of variance followed by Duncan’s multiple range post hoc tests.

## 3. Results

### 3.1. Body Weight, Food Intake, and Water Intake

Table 1 presents body weights and food and water intakes of mice over 6 weeks. At the end of the experiment, the RG group exhibited markedly higher body weights than db/db and HMC groups. The body weights were 42.81 ± 5.08, 47.53 ± 2.50, and 40.43 ± 2.72 g in the db/db, RG, and HMC groups, respectively (*p* < 0.05). Daily food intake was not markedly different among three groups. However, water intake was markedly higher in the db/db group compared to the HMC group. Daily water intake was 3.54 ± 0.44 and 3.16 ± 0.48 mL/day in the db/db and HMC groups, respectively (*p* < 0.05).

### 3.2. Blood Glucose and HbA1c Values

At first, blood glucose levels were not significantly different between the groups (Figure 2). However, after 2 weeks, the fasting blood glucose levels of the HMC group were markedly lower than those of the db/db group. Blood glucose values determined in db/db mice were elevated throughout the experimental period, likely reflecting the progress of diabetes mellitus. However, blood glucose concentration was maintained at a moderate level without an increase in the HMC group. HbA1c values were 11.37 ± 0.78, 6.15 ± 0.63 and 7.24 ± 0.26% in the db/db group, RG, and HMC groups, respectively (Figure 3A). HbA1c values in the HMC group were markedly lower than those in the db/db group (*p* < 0.05).

### 3.3. Plasma Insulin Levels and HOMA-IR

Plasma insulin values were markedly lower in the RG group (132.76 ± 3.03 pmol/L) than those in the db/db group (316.62 ± 6.85 pmol/L) (Figure 3B). Similarly, HMC mice (182.69 ± 2.90 pmol/L) exhibited markedly lowered plasma insulin levels than the mice in the db/db group (*p* < 0.05). HOMA-IR was markedly lower in the HMC group (10.40 ± 0.49) compared to the db/db group (31.35 ± 4.23, *p* < 0.05). The data show that the insulin resistance of the mice in the HMC group was markedly less severe than that of the mice in the db/db group.

### 3.4. Serum Lipid Profiles

TC levels were 214.01 ± 4.92, 220.84 ± 3.48 and 182.79 ± 5.33 in the db/db, RG, and HMC groups, respectively (*p* < 0.05) (Figure 4). TG levels were 236.43 ± 4.38, 216.87 ± 1.35, and 184.27 ± 3.33 in the db/db, RG, and HMC groups, respectively (*p* < 0.05). LDL-C levels were 133.52 ± 8.40, 127.80 ± 4.80, and 82.28 ± 11.45 in the db/db, RG, and HMC groups, respectively (*p* < 0.05). HDL-C levels in the HMC group were the highest among the three groups. HDL-C levels were 33.20 ± 3.85, 49.67 ± 2.88, and 63.66 ± 5.47 in the db/db, RG, and HMC groups, respectively (*p* < 0.05). The serum lipid profiles of the HMC group showed an improvement compared to those of the db/db and RG groups.

### 3.5. Expression of Insulin Signaling Pathway

To clarify whether HMC supplementation promoted the activation of the insulin signaling pathway and led to glucose uptake into skeletal muscle cells, insulin signaling was investigated. As presented in Figure 5, the expression levels of phosphorylated IRS-1 and Akt, and activated PI3K level in mice from the HMC group were markedly increased compared to those in mice in the db/db group. There was a 1.58-fold increase in pIRS-1^tyr612^ levels in the HMC group compared to the db/db group. Similarly, the PI3K activation level was 1.76-fold higher in the HMC group than in the db/db group, and the phosphorylation level of Akt^ser473^ (1.64-fold) was also elevated by HMC supplementation.

### 3.6. Expression of AMPK

The HMC group showed markedly upregulated AMPK^Thr172^ phosphorylation (1.81-fold) in comparison to the db/db-group (Figure 6A). The phosphorylation of acetyl Co-A carboxylase at serine 79 (ACC^Ser79^) was 1.73-fold higher with HMC. These results indicate that HMC promoted the activation of AMPK^Thr172^.

### 3.7. Expression of PM-GLUT4

We investigated the effect of HMC on plasma membrane GLUT4 expression. Figure 6B presents the level of PM-GLUT4 expression in the HMC group compared with that in the db/db group. In the HMC mice, the expression level of PM-GLUT4 was markedly increased, to a level that was 1.92-fold higher than its expression value in db/db mice. These data indicate that HMC can stimulate GLUT4 translocation to plasma membrane.

### 3.8. Expression of AMPK, PPARα, SREBP-1c, and FAS

To identify the molecular mechanisms underlying the ability of HMC to improve dyslipidemia, we investigated the expression of AMPK, the lipogenesis-related transcription factors PPARα and SREBP-1c, and the lipogenesis-related enzyme FAS, in the liver. HMC markedly increased the phosphorylation of AMPK and PPARα (210.98% and 229.43%, respectively) and markedly reduced the expression levels of SREBP-1c and FAS (52.03% and 43.47%, respectively) in HMC mice compared to db/db mice (Figure 7).

## 4. Discussion

Long-term high blood glucose and dyslipidemia can cause grave health problems in human organs or tissues [25,26]. So, the control of high blood glucose and lipid levels is one of the most vital aspects of T2DM management. The db/db mice were chosen as a model system through which to evaluate the effect of HMC on hyperglycemia and dyslipidemia. db/db mice exhibit clinical characteristics similar to those of type 2 diabetic patients, such as hyperglycemia, insulin resistance, and hyperinsulinemia [27].

In the condition of diabetes, glucose cannot effectively flow into the cells, leading to raised blood glucose values and glucose expulsion via urine, which can cause polyuria, polydipsia and polyphagia [28]. The body weight of RG group mice increased the most in comparison to the other groups. Such body weight gain was associated with a raised adipocyte differentiation via activating PPAR-γ, which is a known adverse effect of rosiglitazone [29]. However, supplementing with HMC did not result in similar side effects. Instead, supplementation with HMC alleviated diabetic symptoms, which were evident in the form of reduced water consumption and weight gain.

Fasting blood glucose can be used as an indicator of impaired glucose uptake. When the blood glucose concentration is above a certain level, insulin stimulates glycogen synthesis, and glucose is stored in the liver as glycogen. Muscles also promote glucose uptake to reduce blood glucose levels. However, hyperglycemia occurs if muscle cells fail to absorb glucose due to insulin resistance. Hyperglycemia due to insulin resistance causes increased secretion of insulin, which increases energy requirements and upregulates gluconeogenesis in the liver [30,31]. This indicates that the ability to regulate glucose metabolism is weakened in the liver and muscle due to hyperglycemia caused by insulin resistance. In mice supplemented with HMC, the fasting blood glucose values were generally maintained at moderate levels compared to the db/db group, which exhibited increased levels. Thus, HMC supplementation seemed to improve fasting blood glucose in db/db diabetic mice. HMC is sappanin-type homoisoflavonoid. It is an unusual flavonoid subclass that includes carbon atoms. Brazilin, a homoisoflavonoid, diminishes blood glucose values in diabetic mice [32]. Homoisoflavonoids, isolated from *Polygonatum odoratum*, alleviate high blood glucose levels caused by the impaired uptake of glucose in diabetic mice [33].

HbA1c is of significant value for diabetic patients [34] and is more stable than fasting blood glucose because it reflects the average blood glucose level over 2–3 months. The normal level of HbA1c is 4.0–5.5%, and HbA1c increases by 2–3 times in the presence of diabetes [35]. Elevated HbA1c levels decrease oxygen transport from red blood cells to tissues. High HbA1c levels can cause serious health problems, including heart disease, kidney disease, and nerve damage [36]. Therefore, it is important to reduce HbA1c levels in patients with diabetes. In the present study, supplementation with HMC significantly decreased fasting blood glucose and HbA1c levels, suggesting that it may be useful for treating diabetes due to hyperglycemia.

In the early stages of type 2 diabetes, hyperinsulinemia is shown to overcome insulin resistance [37]. In this study, mice in the db/db group had hyperinsulinemia, whereas mice in the HMC and RG groups showed an improvement in hyperinsulinemia. Our analyses suggested that the plasma insulin levels decreased as insulin resistance was relieved by HMC or RG supplementation. To assess insulin resistance, HOMA-IR was calculated using fasting glucose and insulin concentrations. In this study, HOMA-IR was significantly lower in HMC mice compared to db/db mice. These observations imply that HMC supplementation may contribute to improving insulin resistance.

The improvement in insulin resistance promotes glucose uptake into cells, thereby reducing hyperglycemia. Glucose uptake, a particularly important target in the treatment of type 2 diabetes, is mediated by insulin signaling and the AMPK pathway [38]. Western blotting was conducted to examine the levels of gene expression associated with the insulin signaling pathway. Phosphorylated IRS-1^Tyr612^ binds to and activates PI3K, which in turn phosphorylates Akt^Ser473^. As a result, GLUT4 is transferred to the plasma membrane to facilitate the absorption of blood glucose into the cell [39,40]. In this study, supplementation with HMC significantly increased the expression levels of genes involved in the insulin signaling pathway. In addition, the level of plasma membrane GLUT4 expression was significantly increased in skeletal muscle after HMC supplementation.

According to previous studies, brazilin, a homoisoflavonoid, increased glucose uptake, which was blocked by wortmannin, an inhibitor of PI3K. This study suggested that brazilin might increase glucose uptake by promoting the translocation of GLUT4 to the plasma membrane via the activation of PI3K [41]. This effect was caused by the presence of the four hydroxy groups in this compound. Homoisoflavonoids with multiple hydroxy groups and methoxy groups tend to have much stronger effects on the stimulation of glucose uptake by promoting the translocation of GLUT4 to the plasma membrane than homoisoflavonoids with only one hydroxy group [42]. Thus, we assumed that these two hydroxy groups and one methoxy group in HMC might contribute to the increased activation of the insulin signaling pathway and an elevated plasma membrane GLUT4 expression level in the skeletal muscle of diabetic mice.

AMPK plays a central role in insulin-independent signaling. The AMPK-related glucose uptake pathway in skeletal muscle plays an important role in maintaining glucose homeostasis [43]. The activation of AMPK stimulates the translocation of GLUT4, increases glucose uptake in skeletal muscle, and reduces blood glucose levels [44,45,46,47]. The phosphorylation level of AMPK was elevated in the db/db-HMC group compared to the db/db group. HMC supplementation also seemed to promote glucose uptake via AMPK activation in the skeletal muscles of type 2 diabetic mice.

The metabolic profile of T2DM is characterized by hyperglycemia, which is frequently associated with dyslipidemia. Dyslipidemia is common in patients with type 2 diabetes, and abnormal lipid metabolism is often observed together with increased levels of TG, TC, and LDL-cholesterol, and reduced HDL-cholesterol levels [48]. Decreased plasma LDL-cholesterol levels increase HDL-cholesterol levels and play a role in inhibiting or delaying the progression to diabetic atherosclerosis [49].

In this study, HMC showed potent antidyslipidemic effects in db/db mice. Meanwhile, rosiglitazone, the positive control agent, significantly elevated plasma levels in TG, TC, and LDL-cholesterol. The TZD class comprises antidiabetic drugs that are widely used to improve insulin resistance and control blood glucose in diabetic patients [50]; however, these drugs may cause side effects [51]. In particular, rosiglitazone stimulates adipocyte differentiation, thereby leading to increased fat mass and decreased physical activity [29]. TG, TC, and LDL-cholesterol levels were elevated in the RG group. HMC supplementation improved dyslipidemia without causing side effects such as weight gain. In addition, HMC decreased lipogenesis by regulating the expression of key lipid metabolism genes.

According to a study by Oh et al., hydroxyl groups were shown to markedly downregulate the expression of SREBP-1c and suppress the expression of FAS [52]. In addition, isorhamnetin (a flavonoid with several hydroxyl groups) downregulated the expression of SREBP-1c [53,54]. Collectively, hydroxy and methoxy groups within the flavonoids are critical for the amelioration of dyslipidemia [55]. HMC possesses both methoxy and hydroxy groups bound to its C-16 skeleton. Therefore, we assume that the two hydroxyl groups and one methoxy group in the HMC structure contributed to ameliorating dyslipidemia through the regulation of lipogenic factors in the liver of type 2 diabetic mice.

In conclusion, HMC reduces hyperglycemia by improving insulin resistance and ameliorates dyslipidemia in C57BL/Ksj-db/db mice. Thus, the present study suggests that HMC has the potential to be used as a functional material to management of type 2 diabetes.

## Figures and Tables

**Figure 1 nutrients-14-01951-f001:**
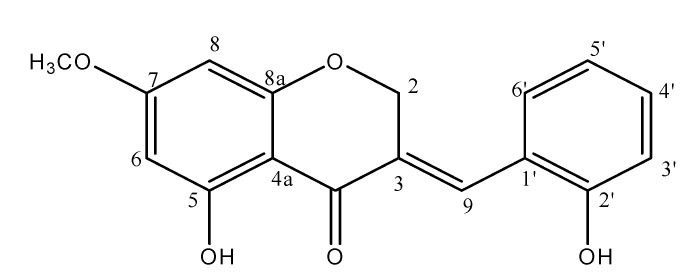
HM-chromanone of chemical structure.

**Figure 2 nutrients-14-01951-f002:**
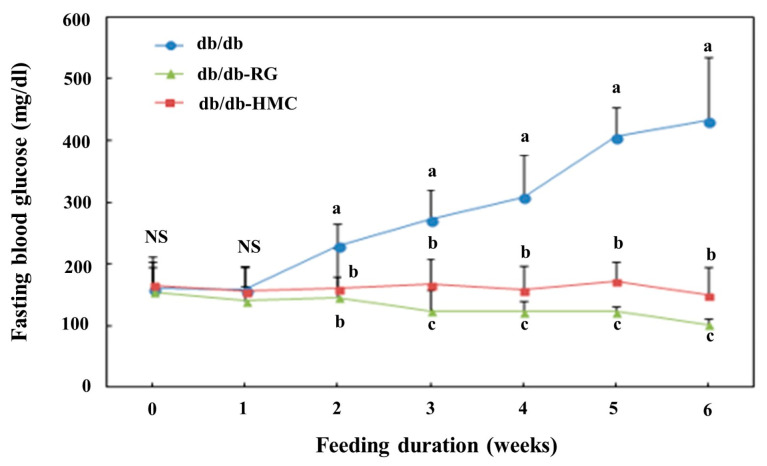
Effects of HM-chromanone administration on fasting blood glucose. Mean ± SD (*n* = 8). ^a–c^ The mean values that are not indicated by a common letter are significantly different among the groups (*p* < 0.05) according to Duncan’s multiple range test. NS, not significant; SD, standard deviation. db/db, db/db-RG (5 mg/kg BW per day) and db/db-HMC group (30 mg/kg BW per day).

**Figure 3 nutrients-14-01951-f003:**
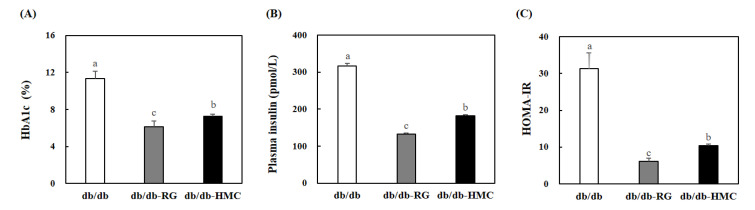
The effects of HMC administration on the levels of HbA1c, plasma insulin and HOMA-IR in C57bL/ksj-db/db mice. (**A**) HbA1c level, (**B**) plasma insulin level, (**C**) HOMA-IR level. Mean ± SD (*n* = 8). ^a–c^ The mean values that are not indicated by a common letter are significantly different among the groups (*p* < 0.05) according to Duncan’s multiple range test. SD, standard deviation. db/db, db/db-RG (5 mg/kg BW per day) and db/db-HMC group (30 mg/kg BW per day).

**Figure 4 nutrients-14-01951-f004:**
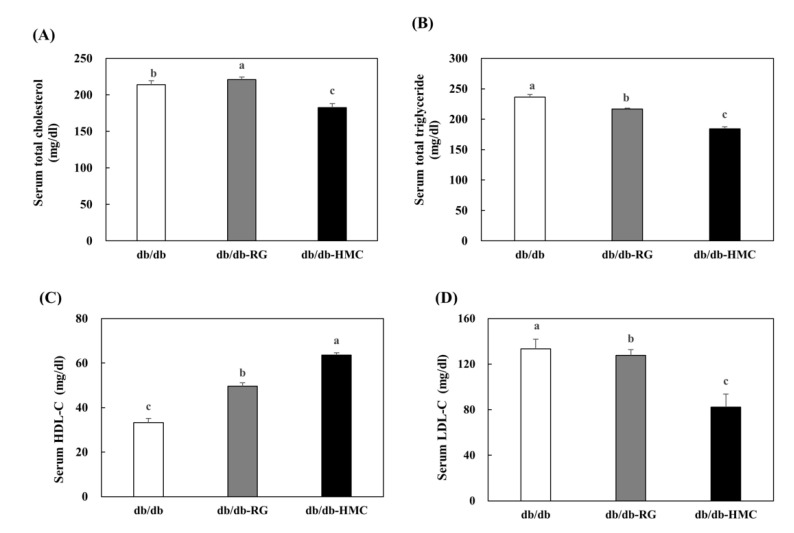
The effects of HMC administration on the levels of serum lipid in C57bL/ksj-db/db mice. (**A**) Serum total cholesterol level, (**B**) Serum total triglyceride level, (**C**) Serum HDL-cholesterol level, and (**D**) Serum LDL-cholesterol level. Mean ± SD (*n* = 8). ^a–c^ The mean values that are not indicated by a common letter are significantly different among the groups (*p* < 0.05) according to Duncan’s multiple range test. SD, standard deviation. db/db, db/db-RG (5 mg/kg BW per day) and db/db-HMC group (30 mg/kg BW per day).

**Figure 5 nutrients-14-01951-f005:**
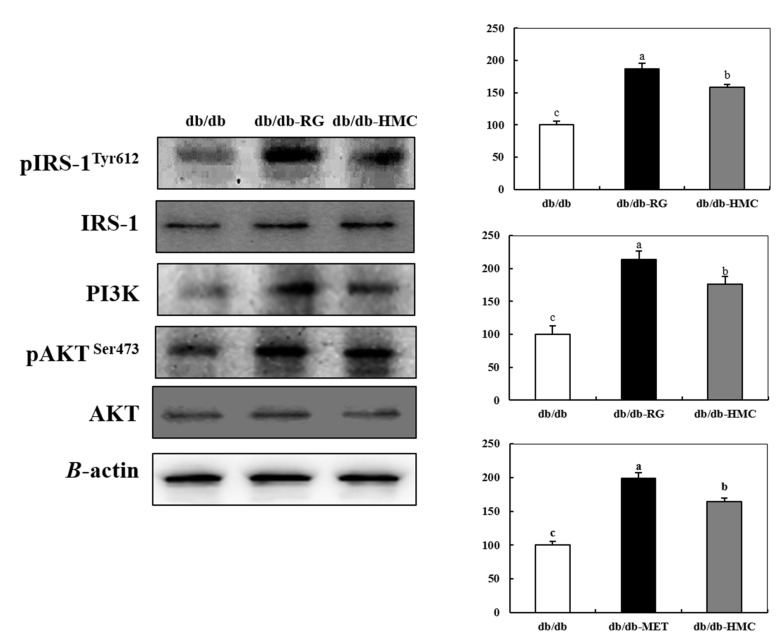
The effects of HMC administration on pIRS-1^Tyr612^, PI3K, and pAKT^Ser473^ expression in skeletal muscle of C57BL/KsJ db/db mice. Mean ± SD (*n* = 8). ^a–c^ The mean values that are not indicated by a common letter are significantly different among the groups (*p* < 0.05) according to Duncan’s multiple range test. SD, standard deviation. db/db, db/db-RG (5 mg/kg BW per day) and db/db-HMC group (30 mg/kg BW per day).

**Figure 6 nutrients-14-01951-f006:**
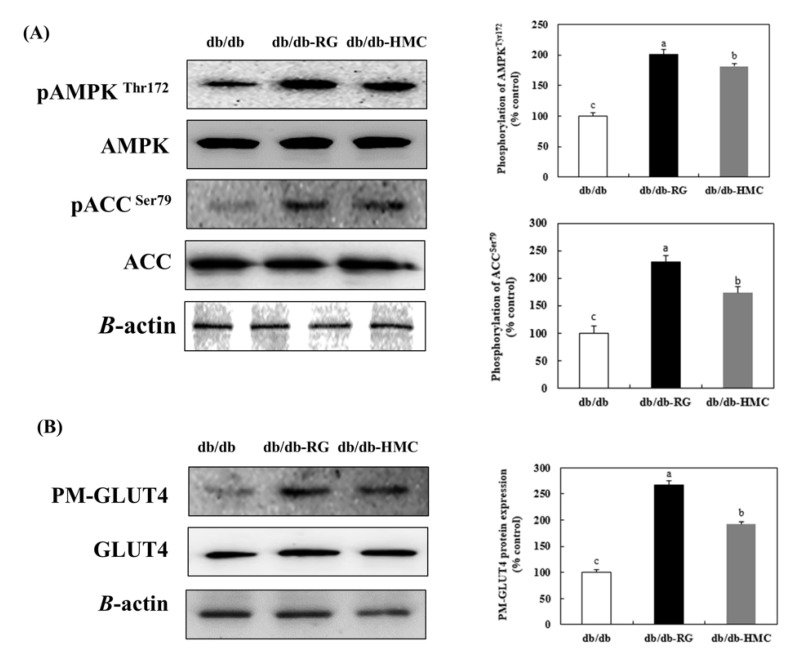
The effects of HMC administration on pAMPK^Thr172^, pACC^Ser79^ and PM-GLUT4 expressions in skeletal muscle of C57BL/KsJ db/db mice. (**A**) Expression of AMPK pathway and (**B**) Expression of PM-GLUT4. Mean ± SD (*n* = 8). ^a–c^ The mean values that are not indicated by a common letter are significantly different among the groups (*p* < 0.05) according to Duncan’s multiple range test. SD, standard deviation. db/db, db/db-RG (5 mg/kg BW per day) and db/db-HMC group (30 mg/kg BW per day).

**Figure 7 nutrients-14-01951-f007:**
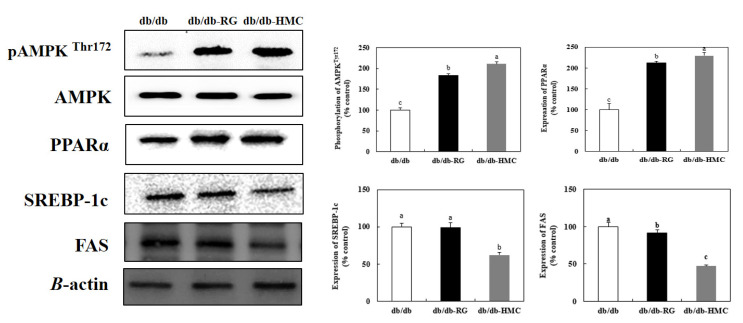
The effects of HMC administration on pAMPK^Thr172^, SREBP-1c, FAS and PPARα expression in liver of C57BL/KsJ db/db mice. Mean ± SD (*n* = 8). ^a–c^ The mean values that are not indicated by a common letter are significantly different among the groups (*p* < 0.05) according to Duncan’s multiple range test. SD, standard deviation. db/db, db/db-RG (5 mg/kg BW per day) and db/db-HMC groups (30 mg/kg BW per day).

**Table 1 nutrients-14-01951-t001:** The effects of HMC administration on body weight, food and water consumption of C57bl/ksj-db/db mice.

	db/db	db/db-RG	db/db-HMC
Water consumption (mL/day)	3.54 ± 0.44 ^a^	1.99 ± 0.26 ^c^	3.16 ± 0.48 ^b^
Food consumption (g/day)	2.35 ± 0.28 ^NS^	2.17 ± 0.29	2.30 ± 0.28
Body weight (g)			
Initial	25.71 ± 1.56 ^NS^	26.54 ± 1.51	26.56 ± 1.28
Final	42.81 ± 5.08 ^b^	47.53 ± 2.50 ^a^	40.43 ± 2.72 ^c^
Body weight gain (g/day)	0.40 ± 0.08 ^b^	0.49 ± 0.02 ^a^	0.33 ± 0.03 ^c^

Mean ± SD (*n* = 8). ^a–c^ The mean values that are not indicated by a common letter are significantly different among the groups (*p* < 0.05) according to Duncan’s multiple range test. ^NS^, not significant; SD, standard deviation; db/db, db/db-RG (5 mg/kg BW per day) and db/db-HMC group (30 mg/kg BW per day).

## Data Availability

Not applicable.

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
