# Peer review of "HM-Chromanone Ameliorates Hyperglycemia and Dyslipidemia in Type 2 Diabetic Mice"

_nutrients, 2022, doi:10.3390/nu14091951_

Round 1
Reviewer 1 Report
This study investigated the effects of (E)-5-hydroxy-7-methoxy-3-(2-hydroxybenzyl)-4-chromanone (HMC) on hyperglycemia and dyslipidemia and related signaling pathways in diabetic C57BL/Ksj-db/db mice. After 6 weeks of treatment with HMC, the glucose homeostatic markers and lipid profiles were improved concurrently with changes in protein expression of genes involved in insulin resistance and lipid metabolism in db/db mice. There are several major issues in study design, statistical analysis method, data interpretation that require comments from the authors.
Material and methods
- What was the solvent for HMC? If the solvent was not distilled water, then another control group with solvent administration should be included in the experiment.
- Please provide the age of mice when the experiment started.
- Please give more details on the dosage of HMC. Please state the theoretical basis of the dose of hormone intervention. Is there any evidence from previous literature? What is the equivalent dosage of HMC for an individual as 30 mg/kg BW used for mice in the study? And what is the amount of Portulaca oleracea one needs to consume to get that dose of HMC?
- Statistical analysis: Have you performed normality test prior to the use of one-way ANOVA? Sometimes glucose and insulin data may not be normally distributed.
- Was p < 0.05 represented as statistical significance? If so, please state in the corresponding section.
- Overall the results were not written in details.
Results
- The data of blood glucose levels as presented in Figure 2 should be analyzed using two-way ANOVA rather than one-way ANOVA, as it should examine the effects of both time and treatment and time*treatment interaction.
- Please provide the statistical analysis results for the comparisons of among groups in the text of results. Otherwise, it is difficult for readers to understand whether the differences among groups are statistically significant or not.
- The last sentence in section 3.7, it is unreliable to state that HMC supplementation increases glucose uptake into cells because the authors did not examine the glucose uptake of skeletal muscle. Upregulation of GLUT-4 did not equal to increase in glucose uptake.
Discussion& Conclusion
- The discussion should be reorganized to avoid too much summarizing background information and repeating the results.
- The final conclusion of “supplementation with HMC may ameliorate hyperglycemia and dyslipidemia in humans” was over interpretive due to the nature of animal study. Same issue with the last sentence in abstract.
Tables/Figures
- Please increase the resolution of figures as all of them look obscured.
- Font size of Figure 7 should be uniform.
Author Response
Point 1: Moderate English changes required
→ Response:
We grammatically re-edited this manuscript by the native speaker "Editage".
Material and methods
- What was the solvent for HMC? If the solvent was not distilled water, then another control group with solvent administration should be included in the experiment.
→ Response:
RG and HMC were dissolved in distilled water (line 84).
- Please provide the age of mice when the experiment started.
→ Response:
After purchasing male C57BL/KsJ-db/db mice (5 weeks), it was acclimatized for 2 weeks, and the experiment was started at 7-week-old.
(line 80) Male C57BL/KsJ-db/db mice (5 weeks) were purchased from JOONG AH BIO (Suwon-si, Gyeonggi-do, Korea) and the db/db mice were fed a pelletized commercial chow diet for 2 weeks after arrival. Before the experiment (at 7-week-old), the mice were randomly divided into three groups (n = 8 per group): db/db, db/db-RG, and db/db-HMC groups.
- Please give more details on the dosage of HMC. Please state the theoretical basis of the dose of hormone intervention. Is there any evidence from previous literature? What is the equivalent dosage of HMC for an individual as 30 mg/kg BW used for mice in the study? And what is the amount of Portulaca oleracea one needs to consume to get that dose of HMC?
→ Response:
①The db/db-HMC group mice were orally administered HMC (30 mg/kg body weight per day, respectively) by gavage, once per day, for 6 weeks.
② As a result of administration of 10mg and 30mg HMC in a preliminary study, it was confirmed that the 30mg HMC administration group had a higher blood glucose lowering effect. Therefore, based on the results, the dosage of 30 mg HMC was determined.
③ For humans, the dose is 1.8g/day/60kg. However, as a result of obtaining the value through the safety factor, it is safe to take 1.8g/day/60kg ÷ 100 = 0.018g/day/60kg.
④ Portulaca oleracea (300g) contains about 2.9mg of HMC.
Assuming 60kg per humans,
|
HMC dose |
Portulaca oleracea intake |
|
30mg/B.W |
1.86kg |
- Statistical analysis: Have you performed normality test prior to the use of one-way ANOVA? Sometimes glucose and insulin data may not be normally distributed.
→ Response: Normality tests were performed before using one-way ANOVA.
- Was p < 0.05 represented as statistical significance? If so, please state in the corresponding section.
→ Response: P < 0.05 was represented in figure legends of each section.
- Overall the results were not written in details.
→ Response: The numerical value of the result was added and explained more detail.
Results
- The data of blood glucose levels as presented in Figure 2 should be analyzed using two-way ANOVA rather than one-way ANOVA, as it should examine the effects of both time and treatment and time*treatment interaction.
→ Response: One-way ANOVA is also used to analyze blood glucose levels. The following is references using the one-way ANOVA as in this study. Please understand this.
- References
- Horakova O, Kroupova P, Bardova K, Buresova J, Janovska P, Kopecky J, Rossmeisl M. Metformin acutely lowers blood glucose levels by inhibition of intestinal glucose transport. Sci Rep. 2019 Apr 16;9(1):6156.
- Choi J, Kim KJ, Koh EJ, Lee BY. Gelidium elegansExtract Ameliorates Type 2 Diabetes via Regulation of MAPK and PI3K/Akt Signaling. Nutrients. 2018 Jan 6;10(1):51.
- Wang Z, Hwang SH, Lee SY, Lim SS. Fermentation of purple Jerusalem artichoke extract to improve the α-glucosidase inhibitory effect in vitro and ameliorate blood glucose in db/db mice. Nutr Res Pract. 2016 Jun;10(3):282-7.
- Seo YJ, Lee K, Chei S, Jeon YJ, Lee BY. Ishige okamuraeExtract Ameliorates the Hyperglycemia and Body Weight Gain of db/db Mice through Regulation of the PI3K/Akt Pathway and Thermogenic Factors by FGF21. Mar Drugs. 2019;17(7):407.
- Please provide the statistical analysis results for the comparisons of among groups in the text of results. Otherwise, it is difficult for readers to understand whether the differences among groups are statistically significant or not.
→ Response: Statistical analysis results for comparison among groups are described.
a-cThe mean values that are not indicated by a common letter are significantly different among the groups (P<0.05) according to Duncan’s multiple range test.
- The last sentence in section 3.7, it is unreliable to state that HMC supplementation increases glucose uptake into cells because the authors did not examine the glucose uptake of skeletal muscle. Upregulation of GLUT-4 did not equal to increase in glucose uptake.
→ Response: I amended the last sentenc in section 3.7.
Discussion& Conclusion
- The discussion should be reorganized to avoid too much summarizing background information and repeating the results.
→ Response: Background information has been added, and discussion has been corrected by removing repetition of results.
- The final conclusion of “supplementation with HMC may ameliorate hyperglycemia and dyslipidemia in humans” was over interpretive due to the nature of animal study. Same issue with the last sentence in abstract.
→ Response: I amended the last sentence in abstracts and final conclusion.
*Abstract: Therefore, these results suggest that HMC could ameliorate insulin resistance, hyperglycemia, and dyslipidemia in type 2 diabetes mice.
*Final conclusion: (line 425) Thus, the present study suggests that HMC has the potential to be used as a functional or medicinal material for the management of type 2 diabetes.
Tables/Figures
- Please increase the resolution of figures as all of them look obscured.
→ Response: I amended the resolution of the figures.
- Font size of Figure 7 should be uniform.
→ Response: Font size of Figure 7 has been modified to be the same.
Reviewer 2 Report
Dear authors.
You presented an interesting work related to the effects of HMC on hyperglycemia and dyslipidemia in diabetic mice. You have taken up the very topical topic of insulin resistance and diabetes therapy, as new therapeutic approaches are still being sought. Below I present my comments on the work.
- Abstract. Line 14 to 17 - if it is possible, you should develop shortcuts IRS-1, Akt, PI3K, GLUT4, AMPK, SREBP-1c and FAS.
- Introduction. Line 68 - please provide information on, why you chose rosiglitazone from your oral hypoglycaemic agents, than for example metformin.
- Materials and methods. Line 86 - what does BW mean - please explain the abbreviation
- Results. Line 214 - it should be standardized whether you write at work about mice with diabetes or mice with type 2 diabetes.
- Discussion. Line 291 - it is worth mentioning the likely mechanism of weight gain with rosiglitazone.
Author Response
Response to Reviewer 2 Comments
Point 1: English language and style are fine/minor spell check required
→ Response:
We grammatically re-edited this manuscript by the native speaker "Editage".
- Abstract. Line 14 to 17 - if it is possible, you should develop shortcutsIRS-1, Akt, PI3K, GLUT4, AMPK, SREBP-1c and FAS.
→ Response 1: If it is possible, I will check the shortcut function and apply it.
- Introduction. Line 68 - please provide information on, why you chose rosiglitazone from your oral hypoglycaemic agents, than for example metformin.
→ Response 2:
The thiazolidinedione rosiglitazone and the biguanide metformin are effective antihyperglycaemic agents with different modes of action; rosiglitazone primarily increases insulin sensitivity, whereas metformin primarily reduces hepatic glucose output.
Rosiglitazone was re-approved by the US Food and Drug Administration, targeting insulin resistance by binding to the transcription factor peroxisome proliferator-activated receptor-γ. In contrast, metformin promotes glucose lowering by reducing hepatic glucose production and gluconeogenesis and by enhancing peripheral glucose uptake.
That is, a suitable positive control for improving insulin resistance in relation to hyperglycemia and dyslipidemia is rosiglitazone. The following is references using the positive control (rosiglitazone) as in this study.
- References
- Xie X, Chen W, Zhang N, et al. Selective Tissue Distribution Mediates Tissue-Dependent PPARγ Activation and Insulin Sensitization by INT131, a Selective PPARγ Modulator. Front Pharmacol. 2017;8:317.
- Min KH, Kim HJ, Jeon YJ, Han JS. Ishige okamurae ameliorates hyperglycemia and insulin resistance in C57BL/KsJ-db/db mice. Diabetes Res Clin Pract. 2011;93(1):70-76.
- Wang C, Jin X, Jin Q, Shi Y, Zhong B and Niu J. Hypoglycemic and Hypolipidemic Effects of a Novel Pan-Peroxisome Proliferator-Activated Receptor Agonist, MBT1805, in db/db Mice. Austin Hepatol. 2021; 6(1): 1016.
- Materials and methods. Line 86 - what does BW mean - please explain the abbreviation
→ Response 3:
BW: body weight.
I amended abbreviations in line 86.
- Results. Line 214 - it should be standardized whether you write at work about mice with diabetes or mice with type 2 diabetes.
→ Response 4:
I amended line 224 about ‘mice with type 2 diabetes’
- Discussion. Line 291 - it is worth mentioning the likely mechanism of weight gain with rosiglitazone.
→ Response 5:
I amended discussion in line 296.
The body weight of mice in the db/db-RG group increased the most in comparison to other groups. Such body weight gain is associated with increased adipocyte differentiation via activation of PPAR-γ, which is a known side effect of rosiglitazone. Fortunately, supplementation with HMC did not result in similar side effects. Rather, supplementation with HMC alleviated diabetic symptoms, which were evident in the form of reduced water consumption and weight gain compared to the db/db and db/db-RG groups.

Round 2
Reviewer 1 Report
The authors have addressed all of my comments. I have no further comments.